# Melatonin as a Chemical Substance or as Phytomelatonin Rich-Extracts for Use as Plant Protector and/or Biostimulant in Accordance with EC Legislation

**Marino B. Arnao \*** and **Josefa Hernández-Ruiz**

Department of Plant Physiology, University of Murcia, 30100-Murcia, Spain; jhruiz@um.es
\* Correspondence: marino@um.es; Tel.: +34 868887001

**Abstract:** Melatonin (*N*-acetyl-5-methoxytryptamine) is a ubiquitous molecule present in animals and plants, and also in bacteria and fungi. In plants, it has an important regulatory and protective role in the face of different stress situations in which it can be involved, mainly due to its immobility. Both in the presence of biotic and abiotic stressors, melatonin exerts protective action in which, through significant changes in gene expression, it activates a stress tolerance response. Its anti-stress role, along with other outstanding functions, suggests its possible use in active agricultural management. This review establishes considerations that are necessary for its possible authorization. The particular characteristics of this substance and its categorization as plant biostimulant are discussed, and also the different legal aspects within the framework of the European Community. The advantages and disadvantages are also described of two of its possible applications, as a plant protector or biostimulant, in accordance with legal provisions.

**Keywords:** biostimulant; fertilizer; melatonin; phytomelatonin; plant protector; plant stress

---

## 1. Introduction

Melatonin (*N*-acetyl-5-methoxytryptamine) is a biogenic amine derived from the amino acid tryptophan, which was discovered in 1958 in the cow pineal gland by Lerner and cols. [1]. Two years later, it was detected in humans and its chemical structure was elucidated. This molecule, which was initially only related to changes in the structure of melanocytes in amphibians, fish and reptiles, was soon found to act as a neurohormone in mammals [2,3]. Since its discovery it has become one of the most researched molecules. In animals, it presents a multitude of physiological actions such as a role in the circadian rhythms of several molecules, and its influence on sleep–wake cycles, mood, motor activity and body temperature changes [4–7]. Its influence on food intake and its relationship with metabolic syndrome has also been demonstrated [8–10]. In other more specific situations such as the physiology of the retina, the immune system, sexual behavior and as an anti-cancer effector, melatonin also has a relevant role [11–15]. In addition, interesting and extensive reviews on the role of melatonin in animals and humans can be consulted [16–23].

In 1995, the presence of melatonin in plants was discovered [24–27]. During the following years there was much reluctance on the part of researchers to accept this, since some refused to believe that a neurohormone could be present in plants, and much less that it had any role in their physiology. A key piece was the elucidation of the melatonin biosynthesis route in plants, localized between the mitochondria, chloroplasts and cytoplasm of cells, and which has been studied with great accuracy by K. Back and J. Kong in rice and *Arabidopsis* plants [28–30]. However, it is now fully accepted that melatonin is present in all plant species and that it presents a panoply of interesting actions. Indeed,

several studies have demonstrated its role in processes such as seed germination, growth and the development of seedlings, leaves and roots. It takes part in organogenesis processes such as rooting and fruiting, and in processes of leaf and fruit senescence. It acts as a protector of the photosynthetic and stomatic system, and as a regulator of various enzymes of the metabolism of carbohydrates, lipids, amino acids, nitrogen, sulfur and phosphorus. It also has a role in the secondary metabolism, enhancing the synthesis of flavonoids, anthocyanins, and carotenoids, among others. It regulates its own biosynthesis and that of several plant hormones such as auxin, abscisic acid, gibberellins, cytokinins, ethylene, polyamines, jasmonic acid and salicylic acid [31–39].

Of all the aspects investigated, its protective action against stress situations has been the most researched and about which most is known. Melatonin exerts a protective action, mediated by major changes in gene expression, both against abiotic (cold, heat, drought, waterlogging, salinity, alkalinity, acid rain, chemical contamination by heavy metals, UV radiation) and biotic (bacteria, fungi, virus) stressors. As a result, plants are more tolerant and/or resistant to the negative action of such stressors [31,36,40–43] (see below). The term "biostimulant" was first proposed to denote "materials that, in minute quantities, promote plant growth" by Zhang and Schmidt (1997) [44]. Later, the definition was modified by Kaufman et al. (2007) as: "Biostimulants are materials, other than fertilisers, that promote plant growth when applied in low quantities" [45]. According Du Jardin (2015), the following definition is proposed: "A plant biostimulant is any substance or microorganism applied to plants with the aim to enhance nutrition efficiency, abiotic stress tolerance and/or crop quality traits, regardless of its nutrients content", and extended as "plant biostimulants also designate commercial products containing mixtures of such substances and/or microorganisms" [46]. Under the EC (European Community) regulation: "Plant biostimulants will be EC marked as fertilizing products stimulating plant nutrition processes independently of the products' nutrient content with the sole aim of improving one or more of the following characteristics of the plant and the plant rhizosphere or phyllosphere: Nutrient use efficiency, tolerance to abiotic stress, crop quality, availability of confined nutrients in the soil and rhizosphere, humification and degradation of organic compounds in the soil". Extensive revision works on this topic can be consulted [47,48]. The objective of this work is to provide sufficient data to establish the clear protective role of melatonin against adverse environmental situations, and to discuss the possible global use of melatonin as a biostimulant and/or bioprotective agent. Current legislation of the EC, is taken into account and the advantages and disadvantages of its use in plant crops destined for animal and human consumption are analyzed.

## 2. Melatonin as a Regulator of Plant Stress Physiology

Although there was much evidence in the 1990s that melatonin could exert some role as an antioxidant agent in animal cells and tissues, it was not until 2004 and 2006, in carrot cells and Chinese licorice (*Glycyrrhiza uralensis* Fisch.), that the possible protective role of melatonin in plants was suggested [49–51], although some curious and previous data existed [52]. The initial idea that melatonin in plants, as in animals, could play an important role as an antioxidant was taking shape and results in this regard became ever more plentiful [53–57]. In addition, studies on melatonin as a possible plant regulator were also progressing, especially since the initial studies of Arnao and cols. on the role of melatonin in plant growth and development, and the so-called auxin-like activity [58–64].

It was not until the publication of results on the action of melatonin on changes in gene expression that the extent and potential of melatonin as a regulatory agent of multiple physiological processes in plants became widely known [64–70]. Exceeding previous expectations, melatonin is capable of activating all known molecular stress mechanisms in plants. Thus, gene regulatory factors involved in the response to cold, high temperatures, salinity, drought, chemical toxicity, etc., and also biotic stress, are up-regulated by melatonin [31,38,40,41,43]. Melatonin also regulates the expression of multiple enzymes related to hormonal homeostasis, up-regulating or down-regulating the expression of genes that encode enzymes of the biosynthetic or catabolic pathways of plant hormones including indole-3-acetic acid (auxin), gibberellins such as gibberellin-4 ($GA_4$), cytokinins, abscisic

acid (ABA) and ethylene. It also others regulators such as salicylic acid (SA), jasmonic acid (JA) and polyamines [31–33,35,38,69,71–75]. In general, subjecting plants to a stressful situation—which leads to an increase in endogenous levels of melatonin—or treatment with exogenous melatonin, results in a stress tolerance response mediated by specific stress response factors and changes in the endogenous levels of plant hormones involved in the response [31–36,38,40–43,75–85]. In addition, the recent identification of a melatonin receptor in *Arabidopsis thaliana* has opened new expectations related to its role as a new plant hormone [86]. Figure 1 shows these aspects in a condensed form.

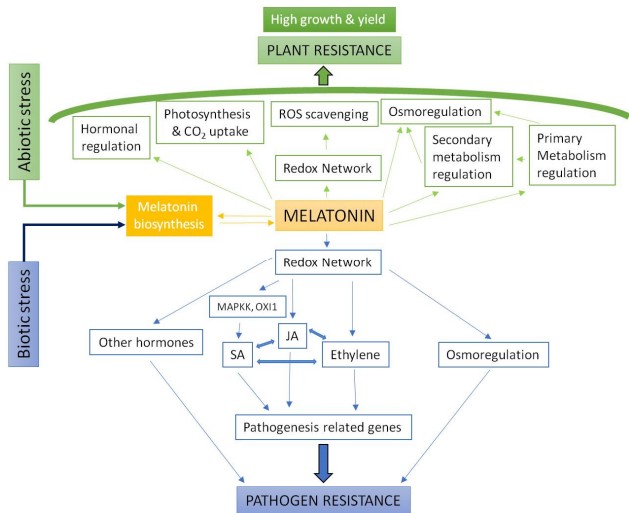

**Figure 1.** Model of redox network/melatonin action on abiotic and biotic stress responses.

## 3. Beneficial Responses to Melatonin Treatments in Different Crops in Stress Situations

Studies conducted with melatonin in plants under both abiotic and biotic stress are numerous. Table 1 compiles many of the studies with an agronomic interest since they deal primarily with crop species for human consumption. Table 1 presents studies classified by plant species, where there are many physiological aspects that are investigated in which melatonin exerts some generally beneficial action. These include seed germination, the growth and vegetative development of plants; photosynthesis, its pigments, photorespiration, stomatic conductance and water economy; the yields of seeds and fruits in adverse conditions; osmoregulation, ion exchange and adjustments in osmotic and hydric potentials, and the regulation of the different metabolisms of carbohydrates, lipids, nitrogen compounds, sulfur and phosphorus cycles. In regards to the secondary metabolism, melatonin induces the biosynthesis of flavonoids, anthocyanins and carotenoids, among others; in hormonal homeostasis, it intervenes in the regulation of all plant hormones and its own biosynthesis. It promotes the rooting process of primary, secondary and adventitious roots while during foliar senescence, melatonin regulates the expression of chlorophyll degradation-related and senescence-induced genes. In the postharvest control of fruits, melatonin increases the ethylene and lycopene content, and regulates many enzymes of the cell wall, ethylene biosynthesis, and primary and secondary metabolisms. It also helps preserve cut flowers; in fruiting it induces parthenocarpy. Finally, its role in bacterial, fungal and viral pathogenic infection should be emphasized, slowing damage and stimulating systemic acquired resistance (SAR) to favor crop health.

Obviously, all the above plant physiology aspects are of interest for application in plant production. Indeed, while many of the above studies were at a laboratory level, others have already been put into practice in crops with excellent results.

In general, exogenous melatonin applications are made through the root system, in irrigation water, or by spraying leaves. In the last case, no adjuvant is needed since melatonin is an amphipathic molecule that crosses biological membranes and the waxy cuticles. Melatonin is transported via the xylem from the roots to the rest of the organs of the plant quite effectively [87,88].

**Table 1.** Studies of different responses to melatonin treatments in different crop species in diverse stress situations.

| Plant Species | Stress Type | Melatonin Treatment (µM) | Effects Observed | Reference |
|---|---|---|---|---|
| Alfalfa | Waterlogging | 100 | ↑ tolerance, growth, photosynthesis, Chls, polyamines, ↓ electrolyte leakage, ROS, ethylene, leaf senescence | [89] |
| | Metal-Cd | 10–200 | ↑ tolerance, growth, Cd transporters, ↓ Cd in roots, ROS | [90] |
| | Oxidative | 1–100 | ↑ lateral root formation, cell division | [91] |
| Apple | Salinity | 0.1 | ↑ shoot height, leaf number, Chls, K$^+$, ↓ electrolyte leakage, ROS | [92,93] |
| | Drought | 100 | ↑ tolerance, re-open stomata, water in leaf, photosynthesis, N uptake, N metabolism, growth, ↓ ABA activity, ROS, leaf senescence | [71,94–96] |
| | Waterlogging | 200 | ↓ chlorosis, wilting of the seedlings, ROS, ↑ tolerance, photosynthesis | [97] |
| | Alkaline | 5 | ↑ tolerance, root system, redox balance, polyamines | [98] |
| | Leaf-senescence | 10 mM | ↓ senescence, ROS burst, ↑ Chls, photosynthesis, sucrose, starch, N | [67,99,100] |
| | *Diplocarpon mali* | 50–500 | ↑ resistance to fungal infection, ↓ leaf lesions, cell death, pathogen expansion | [101] |
| | Apple Replant Disease—ARD | 200 | ↑ growth, photosynthesis, K levels, soil microbial, ↓ ARD effects, ROS | [102] |
| | Apple Stem Grooving Virus | 15 | ↑ shoot regrowth, 95% shoots virus-free, virus-free area | [103] |
| Apricot | - | 10 ppm | ↑ leaf growth, photosynthesis, fruit yield, size and retention, TA, TSS | [104] |
| Banana | Post-harvest | 200–500 | ↑ shelf life of fruits, ↓ ethylene, ripening, quality sharp changes | [105] |
| | Anthracnose | 10 mM | ↑ fruit resistance, banana shelf life, ↓ anthracnose disease | [106] |
| Barley | Cold, drought | 1 mM | ↑ photosynthesis efficiency, ABA, water content, ROS | [72] |
| | Leaf-senescence | 0.01–1 | ↑ Chls, growth, ↓ senescence | [59,107] |
| Bermudagrass | Cold, salt, drought | 20–100 | ↑ growth, osmoregulation, ↓ ROS burst, cell damage | [108,109] |

**Table 1.** *Cont.*

| Plant Species | Stress Type | Melatonin Treatment (µM) | Effects Observed | Reference |
|---|---|---|---|---|
| Broccoli | - | 60 ppm | ↑ growth, photosynthetic attributes: LAI, NAR, AGR, CGR, Chls, carotenoids | [110] |
| Cabbage | Metal-Cu | 1–100 | ↑ germination, growth, ↓ membrane peroxidation | [111] |
| | - | 100–1000 | ↑ growth, anthocyanins, osmoregulation, redox balance, ↓ ABA, senescence factors, Chls degradation | [112,113] |
| Cassava | - | 100 | ↓ ROS, postharvest deterioration, starch degradation | [114] |
| | *Xam* bacterial bligh | | ↑ disease resistance, ↓ bacterial propagation in leaves | [115] |
| Cherry sweet | Rootstocks | 0.5–5 | ↑ number of roots, length, % rooting in 3 cherry rootstocks | [116,117] |
| | | | ↑ photosynthetic pigments, biomass, total carbohydrates and proline | |
| | Orchard trees | 10 | ↓ sweet cherries ripening, anthocyanins | [118] |
| Citrus | Salinity | 1 | ↑ osmoregulation, Chls, ↓ ROS burst, membrane peroxidation | [119] |
| Coffee | Drought | 300 | ↑ tolerance, root system, photosynthesis, gas exchange, $CO_2$ fixation, Chls, ASA-GSH cycle, ↓ ROS, MDA | [120] |
| Cotton | - | 20 | ↑ germination, growth, antioxidant enzymes, $GA_3$, ↓ ROS, MDA, ABA | [121] |
| Cucumber | Cold | 25–500 mM | ↑ germination, ↓ ROS, membrane peroxidation | [122] |
| | Cold | 50–500 | ↑ GSH pool, ↓ ROS | [123] |
| | Cold | 200 | ↑ tolerance, photosynthesis, polyamines, ABA, GSH-ASA cycle, ↓ electrolyte leakage, ROS, | [124,125] |
| | Heat | 100 | ↑ tolerance, N metabol., nitrate, ↓ damage, ammonium | [126] |
| | N-excess | 100 | ↑ tolerance, growth, NPK balance, Ca, ↓ damage, nitrate, ammonium | [127] |
| | Salinity | 1 | ↑ germination, $GA_4$, ↓ ROS, membrane peroxidation, ABA | [69] |
| | | 50–150 | ↑ tolerance, Chls, photosynthesis, GSH-ASA cycle, ↓ ROS | [128] |

**Table 1.** *Cont.*

| Plant Species | Stress Type | Melatonin Treatment (μM) | Effects Observed | Reference |
|---|---|---|---|---|
| | | 1 | ↑ germination, protein biosynthesis, lipid and carbohydrate metabol., TCA, ATP | [129] |
| | Drought | 100 | ↑ germination, root growth | [130] |
| | Oxidative | 50 | ↑ systemic antioxidant defence, GSH, photosynthesis, ↓ ROS | [131] |
| | Metal-Cu | 0.01 | ↑ tolerance, growth, Cu-sequestration, TCA, ATP, GSH, ↓ ROS | [132] |
| | Cinnamic acid | 100 | ↑ tolerance, growth, water and nutrient balance, hormonal balance | [133] |
| Faba bean | Salinity | 100–500 | ↑ plant height, RWC, photosynthetic pigments, osmolites, phenolic | [134] |
| Grape | Drought | 0.05–0.2 | ↑ seedling growth, osmoregulation, photosynthesis, ↓ ROS burst | [135] |
| | Salinity-Rhizobacteria | - | ↑ root growth, RWC, melatonin in roots, colonization, ↓ damage, ROS | [136] |
| | Berry ripening | 100 | ↑ anthocyanins, phenols, flavonoids, proanthocyanidins, resveratrol, ↓ ROS | [137,138] |
| | Berry/Wine | 430 | ↑ size- and ripening-berries, fruity-, spicy- and sweet-wine | [139] |
| Kiwifruit | Drought | 50–200 | ↑ tolerance, photosynthesis, $CO_2$ fixation, growth, biomass, roots, osmoregulation, flavonoids ↓ lipid peroxidation, carotenoid degradation | [140,141] |
| | Heat | 200 | ↑ tolerance, ASA, proline, antioxidant enzymes, ↓ heat damage, ROS | [142] |
| Leek | Cold, heat | 5 | ↑ tolerance, germination, growth | [143] |
| Lychee | Post-harvest | 400 | ↑ redox balance, antioxidant enzymes, ↓ pericarp browning, discoloration, ROS, membrane leakage, loss of phenolics, flavonoids and anthocyanins | [144] |
| Lupin | Several stress | - | ↑ germination, growth, rooting, redox balance, ↓ ROS, foliar senescence | [58,60,62,145] |
| Maize | Salinity, heat | 100 | ↑ photosynthesis, antioxidant enzymes, ↓ ROS, electrolyte leakage | [146,147] |
| | - | 10–1000 | ↑ root and stem growth, plant height, leaf surface area, protein, carbohydrates, Chls | [148] |

**Table 1.** *Cont.*

| Plant Species | Stress Type | Melatonin Treatment (μM) | Effects Observed | Reference |
|---|---|---|---|---|
| | Drought | 100 | ↑ tolerance, growth, photosynthesis, stoma conductance, transpiration, RWC, antiox enzymes, ↓ ROS, MDA | [149–151] |
| | Heat | 10–90 | ↑ tolerance, antioxidant enzymes, osmoregulation, ↓ ROS, MDA, electrolyte leakage | [152] |
| | Metal-Pb | 50–100 | ↑ tolerance, growth, photosynthesis, Chls, RWC, K, Ca levels, ↓ ROS, MDA | [153] |
| | - | 10 | ↑ sugar metabolism, photosynthesis, sucrose phloem loading | [154] |
| | - | 50–500 | protein synthesis, folding, destination and storage, defence, anti-stresses and detoxifying proteins | [155] |
| | Aging seeds | | ↑ viability, growth, antioxidant enzymes, carbohydrate-, secondary-, and amino acid metabol., ↓ ROS, MDA | [156] |
| Melissa (lemon balm) | Metal-Zn-Cd | 1000 | ↑ tolerance, growth, antioxidant enzymes | [157] |
| Mung bean | Cold | 20 | ↑ tolerance, growth, plastids, ↓ ROS, lipid peroxidation | [158] |
| Oat | Salinity, drought | 50–100 | ↑ tolerance, growth, Chls, proline, antioxidant enzymes, ↓ ROS, MDA | [59,159,160] |
| Onion | Cold, heat | 5 | ↑ tolerance, germination, growth | [143] |
| Peach (fruit) | Cold | 50–200 | ↑ juice, TSS, polyamines, GABA, proline, ↓ chilling injury, ROS | [161] |
| | Post-harvest | 100 | ↑ firmness, TSS. ASA, ↓ weight loss, decay incidence, respiration rate, | [162] |
| Pear (tree) | - | 100 | ↑ photosynthesis, fruit size, TSS, sucrose, sorbitol, starch | [163] |
| | Parthenocarpy | 100 | ↑ parthenocarpy with expansion, division mesocarp cells, unviable seeds, GAs | [73] |
| Pear (fruit) | Post-harvest | 100 | ↑ firmness, commercial value, ↓ weight loss, ethylene, softening, core browning | [164,165] |
| Pepper | Salinity, Fe-low | 100 | ↑ growth, Chls, photosynthesis, fruit yield, Fe, K uptake, antioxidant enzymes | [166] |
| | Cold | 1–5 | ↑ germination, growth, antioxidant enzymes, ↓ ROS, MDA | [167] |

**Table 1.** *Cont.*

| Plant Species | Stress Type | Melatonin Treatment (μM) | Effects Observed | Reference |
|---|---|---|---|---|
| | Boron-high | 1 | ↑ tolerance, growth, photosynthesis, ↓ B in leaf and fruit, toxicity | [168] |
| Pea | Oxidative | 50–200 | ↑ photosynthesis efficiency, pigments, water content, ↓ ROS | [169,170] |
| | Metal-Cu | 5 | ↑ plant survival | [171] |
| Plum (fruit) | Cold | 1–1000 | ↑ firmness, postharvest life, ASA, phenols, antioxidant activity, ↓ weight loss | [172] |
| Pomegranate | Cold | 100 | ↑ tolerance, antioxidant enzymes, membrane integrity, phenols, ↓ ROS | [173] |
| Poplar | Oxidative | | ↑ redox balance, proline, ↓ ROS, MDA, membrane damage, electrolyte leakage | [174] |
| Potato | Salinity | 0.1–200 | ↑ tolerance, $K^+$/$Na^+$ homeostasis, ATPase, triacylglycerol breakdown, fatty acid β-oxidation, energy turnover | [175] |
| | *Phytophthora infestans* (potato late blight) | 1–10 mM | ↑ plant innate immunity, fungicide resistance and virulence, synergistic anti-fungal effects of melatonin with fungicides | [176] |
| Radish | Heat | 50–300 | ↑ biomass, quality, antioxidant enzymes, Chls, hormone contents | [177] |
| Rapeseed | Salinity | 0.01–100 | ↑ tolerance, redox balance, ion homeostasis, ↓ ROS, MDA | [178] |
| | Drought | 500 | ↑ tolerance, germination, Chls, stoma size, osmoregulation, antioxidant enzymes, ↓ ROS, MDA | [179] |
| Rice | Cold | 20–100 | ↑ tolerance, growth, photosynthesis, redox balance, ↓ ROS | [180] |
| | Salinity | 10–20 | ↑ Chls, ↓ senescence, ROS, cell death | [181] |
| | Metal-Cd | | ↑ tolerance, growth, photosynthesis, redox balance, panicle number, grain yield | [182,183] |
| | Bacterial blight | 200 | ↓ bacterial proliferation, motility | [184] |
| | Salt, cold, Blast fungus | - | ↑ tolerance, melatonin induction, hormones, ↓ fungi proliferation | [185] |
| | - | 0.5–1 | ↑ seminal roots, lateral roots, root growth, root biomass | [186] |

**Table 1.** *Cont.*

| Plant Species | Stress Type | Melatonin Treatment (μM) | Effects Observed | Reference |
|---|---|---|---|---|
| | Soil | 10–50 | ↑ number of lateral roots, root growth, shaping root architecture | [187] |
| Soybean | Salinity, drought | 50–100 | ↑ tolerance, seedling growth, leaf size, biomass, seed yield | [188] |
| | Drought | 100 | ↑ RWC, Chls, photosynthetic gas-exchange parameters, osmoregulation, antioxidant enzymes, and seed growth-related indicators | [189] |
| | Metal-Al | 0.1–1 | ↑ tolerance, root growth, antioxidant enzymes, osmoregulation, ↓ ROS, | [190] |
| Spinach | Boron | 100–300 | ↑ tolerance, growth, photosynthesis, RWC, $CO_2$ uptake, sugars, carotenoids, redox balance, ↓ ROS, MDA | [191] |
| Strawberry | Post-harvest | 100 | ↑ nutritional quality, antioxidant enzymes, anthocyanins, phenols, GABA, ATP ↓ fungal decay | [192] |
| | Post-harvest | 0.1–1 | ↑ color, firmness, TSS, ASA, flavonoids, ↓ weight loss, senescence, ROS, MDA | [193] |
| Sunflower | Salinity | 15 | ↑ root, hypocotyl growth, antioxidant potential, antioxidant enzymes, GSH | [194–196] |
| Tea plant | Salinity, cold, drought | 100 | ↑ photosynthesis, GSH, ASA, antioxidant enzymes, ↓ ROS, MDA | [197,198] |
| Tobacco | Tobacco mosaic virus | 100 | ↑ tolerance, ↓ virus proliferation, virus-RNA, viral disease | [199] |
| Tomato | Salinity | 50–150 | ↑ photosynthesis, PSII efficiency, D1 protein turnover, ↓ ROS burst | [200] |
| | Salinity | 20–50 | ↑ growth, photosynthesis, Rubisco, proline, C-metabol., ASA-GSH cycle, ↓ ROS, MDA | [201] |
| | Salinity | 150 | ↑ tolerance, photosynthesis, PSII repair, ASA-GSH cycle, ↓ ROS | [202] |
| | Cold | 100 | ↑ antioxidant enzymes, GSA, ASA, $CO_2$ uptake, sucrose, proline, Calvin cycle, polyamines, ↓ ROS, MDA, electrolyte leakage | [203] |
| | Cold | 100 | ↑ tolerance, growth, VAZ cycle, photosynthesis, photosystem efficiency, ↓ ROS, MDA, photoinhibition | [204] |
| | Cold-fruit | 100 | ↑ tolerance, proline, polyamines, membrane integrity | [205] |

**Table 1.** *Cont.*

| Plant Species | Stress Type | Melatonin Treatment (µM) | Effects Observed | Reference |
|---|---|---|---|---|
| | Heat | 10 | ↑ thermotolerance and cell protection | [206,207] |
| | Heat-pollen | 20 | ↑ thermotolerance, polen germination, antioxidant enzymes, reproductive development | [208] |
| | Metal-Cd | 25–500 | ↑ Cd tolerance, phytochelatins, ATPase activity | [209] |
| | Metal-Cd | - | ↑ Cd tolerance, heat-shock factor A1a induction by melatonin | [210] |
| | Metal-Cd-Se | - | ↑ growth, photosynthesis, electrolyte leakage, phytochelatins, GSH, ↓ ROS, Cd leaf, | [211] |
| | Alkalinity | 0.25–1 | seedling growth, photosynthesis, ion homeostasis, burst | [212] |
| | Acid rain | 100 | ↑ tolerance, growth, chloroplast integrity, photosynthesis, antioxidant enzymes, ↓ ROS, MDA | [213] |
| | Drought | 100 | ↑ tolerance, waxes-cutin leaf, RWC, Chls, | [214] |
| | Drought | 100 | ↑ tolerance, Chls, antioxidant enzymes, p-coumaric acid, ↓ ROS, MDA | [215] |
| | S-low | 100 | ↑ S uptake, assimilation, transport and metabolism, peroxiredoxins, redox homeostasis, ↓ ROS, DNA damage | [216] |
| | Rooting | 50 | ↑ adventitious root formation, auxin, auxin transport and signal transduction | [217] |
| | On vine-ripening | | ↑ fruit yield and quality, ASA, citric acid, lycopene, TSS, Ca, P ↓ N, Mg, Cu, Zn, Fe, Mn, | [218] |
| | Post-harvest | 50 | ↑ fruit ripening, fruit quality, colour, carotenoids, polygalacturonase and related, biosynthesis, perception and signalling of ethylene, anthocyanins, ↓ weight loss | [74,219] |
| | Mosaic virus | 100 | ↑ tolerance, ↓ virus proliferation, virus-RNA, viral disease | [199] |
| Valerian | Metal-Zn-Cd | 1000 | ↑ tolerance, growth, antioxidant enzymes | [157] |
| Watermelon | Cold | 150 | ↑ photosynthesis, ↓ cold-related microRNA | [220] |
| | Salinity | 50–500 | ↑ tolerance, growth, photosynthesis, antioxidant enzymes, GSH, ASA, ↓ ROS, MDA | [221] |

**Table 1.** *Cont.*

| Plant Species | Stress Type | Melatonin Treatment (μM) | Effects Observed | Reference |
|---|---|---|---|---|
| | Metal-V | 0.1 | ↑ tolerance, growth, photosynthesis, antioxidant enzymes, ↓ V level, V transport, ROS, MDA | [222] |
| Wheat | Cold | 1000 | ↑ redox balance, Chls, osmoregulation, ↓ ROS | [223] |
| | Cold | 1000 | ↑ tolerance, growth, Chls, photosynthesis, $CO_2$ uptake, grain filled | [224] |
| | Cold | 1000 | ↑ photosynthesis, stomatal conductance, antioxidant enzymes, membrane stability | [225] |
| | Salinity | 1 | ↑ tolerance, growth, photosynthesis, IAA, polyamines, ↓ ROS | [226] |
| | Salinity | 50–500 | ↑ growth, yield, antioxidant enzymes, ↓ ROS, MDA | [227] |
| | Drought | 500 | ↑ tolerance, RWC, photosynthesis, antioxidant enzymes, ASA, GSH, ↓ ROS, membrane damage | [228] |
| | Metal-Cd | 100 | ↑ tolerance, antioxidant enzymes, ASA, GSH, ↓ ROS | [229] |
| | Metal-Cd | 50–100 | ↑ tolerance, growth, Chls, photosynthesis, RWC, Ca, K, antioxidant enzymes, ↓ ROS, MDA, Cd | [230] |
| | Metal-Zn | 1000 | ↑ tolerance, Chls, photosynthesis, Rubisco, ATPase | [231] |
| | N-low | 1 | ↑ N and nitrate, N absorption, N metabolism, growth, yield, in shoots and roots | [232] |

↑, Increased content or increased action. ↓, decreased content or decreased action; ABA, abscisic acid; AGR, absolute growth rate; ASA, ascorbic acid; CGR, crop growth rate; Chls, chlorophylls; CMC, component materials categories of fertilizers; EC, European Community; ECHA, European Chemical Agency; EU, European Union; GA$_4$, gibberellin-4; GABA, γ-aminobutyric acid; GSH, glutathione; JA, Jasmonic acid; LAI, leaf area index; MDA, malondialdehyde; MAPKK, mitogen-activated protein kinase cascade; NAR, net assimilation rate; OXI1, oxidative signal-inducible1 kinases; PFC, product function categories of fertilizers; ROS, reactive oxygen species; RWC, relative water content; SA, salicylic acid; SAR, systemic acquired resistance; TA, total valuable acidity; TCA, Krebs cycle; TSS, total solid soluble.

## 4. Melatonin in the Health and Environment of EC

In accordance with the Classification, Labelling and Packaging (CLP, EC-No 1272/2008) regulation, which is based on the United Nations' Globally Harmonized System, which has a purpose to ensure a high level of protection of health and the environment, as well as the free movement of substances, mixtures and articles, the European Chemical Agency (ECHA) classified melatonin (EC No. 200-797-7 (CAS 73-31-4), *N*-(2-(5-methoxyindol-3-yl)-ethyl)-acetamide), as a non-hazardous substance in terms of physical and chemical hazards. With respect to human health, it is classified as a non-hazardous substance in the oral, dermal, inhalation and irritation categories, and in regards to mutagenicity and carcinogenicity. However, melatonin is classified as a health hazard substance (code H-361) in terms of reproductive toxicity because it is suspected of damaging fertility or an unborn child. This classification reflects one of its multiple functions as an animal hormone, in which its participation in the modulation of sexual behavior in mammals has been demonstrated, and also, it is believed, the same of fertility [233,234]. In fact, it is usually applied to sheep as a hormonal regulator of sexual zeal to homogenize the reproductive process in ovine, with demonstrated higher conception and pregnancy rates when applied [235]. Nevertheless, melatonin is classified as non-hazardous in terms of its possible damage to the environment and atmosphere.

## 5. Melatonin as an Active Substance or as a Plant Biostimulator/Protector in Crops: Concepts and Legal Considerations in EC

After many changes and adaptations, the EC finally seems to have established its policy regarding the authorization, classification, use, distribution, importation, management, etc., of plant protectors and fertilizers, in an attempt to improve agricultural production, while minimizing risks and hazards for humans, animals and the environment. In order to establish the minimum basis for the possible use of melatonin in plant production and post-harvest application, several requirements regarding its human consumption must be taken into account:

(i) Melatonin is a highly studied substance that has given rise to abundant physicochemical and biological data; (ii) there are numerous studies in animals and humans regarding its beneficial effects on health, in aspects as diverse as neurodegenerative, immunological, liver, renal, heart, skin and gastrointestinal diseases, in addition to osteopathy, retinopathy, etc. It also helps in the treatment of various cancers, particularly, chemical and radiological therapies; (iii) in regards to melatonin for human consumption, although it is classified as a drug in the EC, there are some cases in which it does not need a medical prescription, such as those where the amount of melatonin is less than 1 mg. Generally, these are used for jet-lag and sleep disorders. In many other countries (e.g., USA, Canada) melatonin is not treated as a drug, but as a food supplement; (iv) in no case has melatonin been declared as toxic, even at the intake of 1 g/day. Only some slight side effects such as migraine and headache have been described.

The possible use of melatonin in plant production involves particular aspects such as: (i) Melatonin is a molecule that exists in all living things, from bacteria to humans, but also in plants, algae, fungi, etc.; (ii) its action in animals and humans is well known since it has been investigated for many years. In plants, although many physiological effects of melatonin are known, new data are being acquired every day; (iii) in all cases, only positive effects have been described, all beneficial for the development of plants (the same can be said for animals); (iv) little information is available on its effect on bacteria and fungi, especially those that are part of the soil microbiota (rhizosphere); (v) there are also few or no data on its effect on the environment, in particular on agricultural and aquatic fauna; (vi) the levels of melatonin described in plants, and which appear to be effective in pharmacological treatments known to date, are much higher than those described in animals or humans, which may be a cause for caution.

Council Directive 91/414/EEC of 15 July, 1991 concerning the marketing of plant protection products provides rules governing plant protection products and the active substances contained in those products. This old directive has been replaced by two more current ones that are as follows:

- Regulation #1. Regulation EC 1107/2009 of the European Parliament and of the Council of 21 October 2009 concerning the placing of plant protection products on the market and repealing council directives 79/117/EEC and 91/414/EEC, and;
- Regulation #2. Regulation EU 2019/1009 of the European Parliament and of the Council of 5 June 2019 laying down rules on the making available on the market of EU fertilizing products and amending Regulations (EC) No. 1069/2009 and (EC) No. 1107/2009 and repealing Regulation (EC) No 2003/2003.

If we review the actions confirmed so far for melatonin in plants, we find that melatonin exerts a clear action as a plant protector in situations of biotic stress against bacterial, fungal and viral diseases (Regulation #2), but it can also be used as an agent against situations of abiotic stress (Regulation #1). Thus, Regulation #2 says in point 22:

> "Certain substances, mixtures and micro-organisms, referred to as plant biostimulants, are not as such inputs of nutrients, but nevertheless stimulate plants' natural nutrition processes. Where such products aim solely at improving the plants' nutrient use efficiency, tolerance to abiotic stress, quality traits or increasing the availability of confined nutrients in the soil or rhizosphere, they are by nature more similar to fertilising products than to most categories of plant protection products. They act in addition to fertilisers, with the aim of optimising the efficiency of those fertilisers and reducing the nutrient application rates. Such products should therefore be eligible for CE marking under this Regulation and excluded from the scope of Regulation (EC) No 1107/2009".

These two regulations attempt to classify the substances and products applicable to crops into two large groups: Those that are plant protectors (phyto-sanitary) (Regulation #1) and those that can be used as fertilizers (Regulation #2). As we have seen in the previous section, melatonin is classified as a health hazard substance (code H-361) for its reproductive toxicity in ECHA, so its possible authorization as an active substance by regulation EC 1907/2006 of Registration, Evaluation, Authorisation and Restriction of Chemicals (REACH) could be difficult.

Although Regulation #1 on plant protection products extends the concept of an active substance, since it includes microorganisms and preparations (art. 1 point 2): This Regulation shall apply to substances, including micro-organisms having general or specific action against harmful organisms or on plants, parts of plants or plant products, referred to as 'active substances', some interesting restrictions appeared in:

- Art. 23b: "Basic substances shall be approved in accordance with paragraphs 2 to 6. ( . . . ) For the purpose of paragraphs 2 to 6, a basic substance is an active substance which ( . . . ), (b) does not have an inherent capacity to cause endocrine disrupting, neurotoxic or immunotoxic effects";
- Annex II, Impact on Human Health, 3.6.5: "An active substance, safener or synergist shall only be approved if, on the basis of the assessment of community or internationally agreed test guidelines or other available data and information, including a review of the scientific literature, reviewed by the Authority, it is not considered to have endocrine disrupting properties that may cause adverse effect in humans, unless the exposure of humans to that active substance, safener or synergist in a plant protection product, under realistic proposed conditions of use, is negligible, . . . " and in;
- Annex II. Ecotoxicology, 3.8.2. An active substance, safener or synergist shall only be approved if, on the basis of the assessment of community or internationally agreed test guidelines, it is not considered to have endocrine disrupting properties that may cause adverse effects on non-target organisms unless the exposure of non-target organisms to that active substance in a plant protection product under realistic proposed conditions of use is negligible.

Thus, taking into account all this legal information, and ruling out the possibility of using melatonin as an active substance (pure chemical substance) for agronomic application, the possibility of using plant, bacterial, algae, or fungi extracts rich in melatonin would remain. Thus, a good plan

might be to use plant (or other) extracts rich in melatonin as a fertilizer, in the category of biostimulants. A biostimulant could also be defined as a formulated product of biological origin that improves plant productivity as a consequence of the emergent properties of its constituents. Thus, biostimulants could be defined by their demonstrated mode of action and origin, or solely by their demonstrated beneficial impact on plant productivity. The challenges in developing a definition are also complicated by the multi-component and largely undefined composition of many biostimulant products and the possibility that the activity of a biostimulant may not be explained by the presence of any individual constituent, but is a result of the interaction of many constituents in the product. Indeed, most biostimulants in use today are complex mixtures of chemicals derived from a biological process or the extraction of biological materials [236].

According Regulation #2 (EU 2019/1009) on fertilizing products, in Annex I, Product Function Categories (PFCs) of EU fertilizing products, in Category 6, two types of plant biostimulant can be developed: Microbial plant biostimulants (subtype A) and non-microbial plant biostimulants (subtype B). In Annex II, it says: "An EU fertilizing product shall consist solely of component materials complying with the requirements for one or more of the CMCs listed in this Annex", where the different component materials categories (CMC) were defined. Of interest are the following:

- CMC2: Plants, plant parts or plant extracts is described as: "An EU fertilizing product may contain plants, plant parts or plant extracts having undergone no other processing than cutting, grinding, milling, sieving, sifting, centrifugation, pressing, drying, frost treatment, freeze-drying or extraction with water or supercritical $CO_2$ extraction. For the purpose of this point, plants include mushrooms and algae and exclude blue-green algae (cyanobacteria)."
- CMC6: Food industry by-products, point (e): "Plants, plant parts or plant extracts having undergone only heat treatment or heat treatment in addition to processing methods referred to in CMC 2"
- CMC7: Micro-organisms. "An EU fertilising product belonging to PFC 6A may contain micro-organisms, including dead or empty-cell micro-organisms and non-harmful residual elements of the media on which they were produced".

The strategies to obtain melatonin-rich extracts may involve microorganisms (PFC6A) or plants (PFC 6B). At present, there seem to be no data on the production of melatonin by bacteria or fungal cultures. The objective to obtain melatonin-rich plants (CMC2) is ambitious since phytomelatonin levels in plants are usually very low, and less than 5–10 ng per gram of plant. An exhaustive classification of many plants according to their phytomelatonin content can be consulted [37,237,238]. Generally, medicinal plants have high phytomelatonin content, but this tends to vary widely due to the varied origin of plants, technical conditions of growth, variety, post-harvest treatment, etc. Several strategies can be followed: (i) Selecting plant species with high levels of phytomelatonin which can be extracted and concentrated, and (ii) inducing the biosynthesis of phytomelatonin in in vitro cultured pre-selected plant tissues. A discussion on this aspect can be consulted [239]. Our group is developing a formulation where only aromatic/medicinal plants are used to obtain a botanical mixture rich in phytomelatonin through the application of a simple process. A rigorous plant selection protocol and careful management will ensure high phytomelatonin content in the plant extracts generated. The formulation and its protocol are being patented before being made available to interested companies for commercial exploitation. We are currently characterizing it and conducting the appropriate studies and bioassays in plants to confirm its beneficial biological activity related with its high phytomelatonin content.

Figure 2 shows, according to the legislation analyzed, the pros and cons of melatonin (as a chemical substance) and phytomelatonin-rich extracts and its possible regularization as a plant protector or fertilizer (biostimulant).

**Melatonin (active substance) as plant protector**

**PROS**
- Is effective against bacteria, fungi and virus
- Activates systemic immune response
- Activates main energy metabolism
- Activates secondary metabolism
- Is water soluble, fat soluble
- Is relatively cheap

**CONS**
- Legal norms more restrictive
- Classification as a health hazard compound
- Need for studies in soil fungi
- Need for studies in arthropods, etc.
- Need for studies in aquatic environments
- Presents some chemical impurities

**Phytomelatonin (plant extracts) as biostimulant**

**PROS**
- Induces several responses against stressors
- Shows positive response for different abiotic stressors
- Acts as growth and rooting biostimulator
- Inhibits senescence
- Induces hormonal responses
- Is high in antioxidants, flavonoids, etc.
- Can be used in root and leaf treatments
- Natural origin, no chemical by-products

**CONS**
- Search for and selection of plants with high phytomelatonin content
- Need for protocol for extraction, etc.
- Limitations regarding its concentration
- Legal norms less restrictive
- More expensive
- Toxicology to be determined

**Figure 2.** Pros and cons of the possible use of chemical melatonin and rich-phytomelatonin extracts according to EC legislation.

## 6. Future Prospects

Numerous studies with melatonin have resulted in a set of data that indicate the excellent beneficial effects that this compound has on plants, especially in stress situations. It should not be forgotten that melatonin is a natural compound, endogenous to plants and other organisms including humans. It is this last aspect that makes it more interesting and also more delicate or sensitive, when using it as a plant protective agent or as a biostimulator. However, demonstrating through trials that its use is possible in crops and does not entail risks to human and wildlife health will be the only way forward in this field. The alternative of using phytomelatonin-rich extracts seems more interesting, but also more laborious. The search and selection of plants with high endogenous levels of phytomelatonin is a first requirement for subsequent extraction and preparation. The analysis and study of its potential as a protector against plant stress will throw light on the true effect on crops. However, although many aspects of the mechanism of action of phytomelatonin are already known, there are other relevant aspects to study as: (i) The optimal mode of application, time and rate; (ii) the phenological state; (iii) the effect on rhizosphere; (iv) the persistence in soil or in foliar applications; (v) the synergic or antagonic effects with other plant treatments (pesticides, fertilizers, etc.), among others. Obviously, companies in the phytochemical sector (manufacturers) will need to start field studies and deal with possible legal regularization.

## Abbreviations

| | |
|---|---|
| ABA | abscisic acid |
| AGR | absolute growth rate |
| ASA | ascorbic acid |
| CGR | crop growth rate |
| Chls | chlorophylls |
| CMC | component materials categories of fertilizers |
| EC | European Community |
| ECHA | European Chemical Agency |
| EU | European Union |
| $GA_4$ | gibberellin-4 |
| GABA | $\gamma$-aminobutyric acid |

| GSH | glutathione |
|---|---|
| JA | jasmonic acid |
| LAI | leaf area index |
| MDA | malondialdehyde |
| MAPKK | mitogen-activated protein kinase cascade |
| NAR | net assimilation rate |
| OXI1 | oxidative signal-inducible1 kinases |
| PFC | product function categories of fertilizers |
| ROS | reactive oxygen species |
| RWC | relative water content |
| SA | salicylic acid |
| SAR | systemic acquired resistance |
| TA | total valuable acidity |
| TCA | Krebs cycle |
| TSS | total solid soluble |

**Author Contributions:** The manuscript was conceived by M.B.A. and written and revised by M.B.A. and J.H.-R.

**Funding:** Not financial support available for this review.

**Conflicts of Interest:** The authors declare no conflicts of interest.

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
