# Peer review of "Melatonin as a Chemical Substance or as Phytomelatonin Rich-Extracts for Use as Plant Protector and/or Biostimulant in Accordance with EC Legislation"

_agronomy, doi:10.3390/agronomy9100570_

Round 1
Reviewer 1 Report
In the current report the authors have carefully discussed the pros and cons regarding melatonin as a chemical substance or as phytomelatonin rich-extracts for use as plant protector and/or biostimulant in accordance with EC legislation. This is a very interesting article and it provides valuable thoughts on the future prospective of melatonin used in agriculture. As to the physiology of melatonin in plants, much information has been updated. The only ignored information is that the authors have not mentioned or discussed melatonin being mainly synthesized in mitochondria and chloroplasts in plants. This important aspect should be discussed and the related references should be cited. Another minor issue is that some sentences are too long to read and they could be broken to several short sentences.
Author Response
Rev-1
Thank you for your comments. We have incorporated information on melatonin biosynthesis that occurs in mitochondria, chloroplast and cytoplasm of plant cells and three references (lines 47-50) in the revised version. Our English teacher, a great Welshman, has revised and improved the text.
Reviewer 2 Report
The study written by experienced researchers entitled “Melatonin as a chemical substance or as phytomelatonin rich-extracts for use as plant protector and/or biostimulant in accordance with EC legislation“ provides valuable data and represents important contribution to the field with the topic interesting for readers. However, some points must be implemented.
Lines 38-42 are without references; I suggest authors to cite properly the individual mechanisms of action of melatonin (in animals/humans) in the text. References 4-11 quoted in the end of the first paragraph are too general and some of them do not cover above mentioned activities of melatonin in animal cells/tissues. I strongly suggest to use following valuable references for specific mechanisms:
Circadian rhythms:
Xie et al. Neurol Res. 2017 Jun;39(6):559-565 (sleep disorders)
Vadnie et McClung. Neural Plast. 2017;2017:1504507 (mood)
Carpentieri et al. Chronobiol Int. 2015;32(7):994-1004 (motor activity)
Blume et al. Eur J Neurol. 2019 Aug;26(8):1051-1059 (body temperature)
Food intake: Markova et al. Acta Vet Brno 2003; 72:163-73.
www link: https://actavet.vfu.cz/72/2/0163/
Metabolic effects: Bojkova et al. Acta Vet Brno. 2006; 75(1): 21-32
www link: https://actavet.vfu.cz/75/1/0021/
Metabolic syndrome: Cardinali et Hardeland. Neuroendocrinology. 2017;104(4):382-397.
Anti-cancer effects:
Kubatka et al. Folia Biol (Praha) 47, 5-10, 2001 (melatonin significantly decreased tumor volume by almost 70 % vs controls in this study!)
Di Bela et al. Int J Mol Sci. 2013 Jan 24;14(2):2410-30.
Immunomodulatory activities:
Nabavi et al. Crit Rev Food Sci Nutr. 2019;59(sup1):S4-S16.
Carrascal et al. Curr Pharm Des. 2018;24(14):1563-1588.
Moreover, there are several scientifically valuable sentences, however, they are without proper reference, such as line 61, line 81 or line 85, please correct. Moreover, please use references for data mentioned in Figure 1.
Table 1 is excellent and scientifically sound! Conclusion provides the answers to the study aims and also includes limitations.
Author Response
Rev-2
Thank you for your kind comments. We have incorporated the information on animal/human melatonin and all the valuable references suggested. The changes affect the lines 38-44, line 65, line 85 and line 89 of the revised version.